# Microstructure and Mechanical Properties of 34CrNiMo6 Steel Repaired by Friction Stir Processing

**DOI:** 10.3390/ma12020279

**Published:** 2019-01-16

**Authors:** Zhongwen Wu, Chunping Huang, Fencheng Liu, Chun Xia, Liming Ke

**Affiliations:** Engineering Research Center in Additive Manufacturing, Nanchang Hangkong University, Nanchang 330063, China; wuzhongwen_1994@163.com (Z.W.); fencheng999@nchu.edu.cn (F.L.); xiachun2002@163.com (C.X.); liming_ke@126.com (L.K.)

**Keywords:** friction stir processing repair, 34CrNiMo6 steel, microstructure, mechanical properties

## Abstract

Repairing damaged parts using proper repairing methods has become an important means to reduce manufacturing and operational costs and prolong the service life of 34CrNiMo6 steel structures. In the conventional fusion repairing method, welding wire and powder are often used as filling materials. Filling materials are often expensive or difficult to find. Some metallurgical issues (such as solidification crack, higher distortion) were also found with these methods. At the same time, most of the equipment that requires welding wire and powder is expensive. In this study, a new method based on friction stir processing (FSP) was successfully employed to repair 34CrNiMo6 steel, using a block as filling material. Filling blocks are much cheaper than conventional fusion repair consumables. As a result of solid-state repair, this method can also avoid the metallurgical issues of fusion repair. The microstructure and mechanical properties of the repaired samples were investigated using OM (Optical Microscope), SEM, EDS (Energy Dispersive Spectroscopy), XRD, and a Vickers hardness electronic universal tensile tester. The results showed that 34CrNiMo6 steel was successfully repaired by this method, with no defect. Tensile tests showed that the maximum ultimate strength (UTS) was 900 MPa and could reach 91.8% of that of the substrate. The fracture mode of the tensile samples was ductile/brittle mixed fracture. Hence, the repairing method based on FSP appears to be a promising method for repairing castings.

## 1. Introduction

In this study, we focused on 34CrNiMo6 steel, a typical high-strength quenched and tempered steel with medium-carbon low-alloy content, which has been widely used in various industrial fields. Automobile connecting rods, large-scale crankshafts, gear shafts, axles of high-speed railway trains, and forged rotors are the most important applications of this steel [1,2,3]. Such parts often serve under the condition of very strong force and so inevitably encounter service damage. Repairing damaged parts by proper repair methods has become an important means to reduce manufacturing and operational costs, prolonging the service life of 34CrNiMo6 steel structures.

Previous studies have shown that mechanical processing, spraying [4,5], Tungsten Inert Gas Welding (TIG) [6], and Metal Inert Gas welding (MIG) are the most important repair methods at present. Although these methods have many advantages, they also have many disadvantages. The spraying technology involves the mechanical bonding between coating and matrix, which have poor bonding strength, peel easily, and have poor impact resistance. The brush plating technology has low repair efficiency, with poor adhesion between coating and matrix. Both techniques have thin coating layers, making them suitable only for surface repair. In addition to the techniques mentioned above, laser forming repair (LFR) has attracted attention in recent years. Huang et al. [7,8] repaired 34CrNiMo6 substrates with 34CrNiMo6 powder and investigated the resulting microstructure and tribological properties. Yang et al. [9] repaired high-strength steel by LFR and investigated the microstructure and mechanical properties. Zeeshan et al. [10] repaired 30CrMnSiA with a H18CrMoA filler rod and investigated the effect on mechanical and corrosion resistance. Sun et al. [11] investigated the effect of laser cladding on the fatigue behavior of AISI4340 steel. The results showed very poor tensile properties in the clad and Heat affected zone (HAZ). From former studies, it can be found that a welding wire or rod is necessary to fill the defects in the conventional fusion welding repair process. Hot cracking, cold cracking, lack of fusion, and many other defects were found in the post-welding components [12].

However, for some medium-carbon low-alloy steel, these consumable materials are not available or are expensive. To solve this problem, a new repair method based on the friction stir processing (FSP) was studied in this paper. The block which can be obtained from the waste produced by machining was used as repair material. This method, which is derived from FSP, can be widely used in repair fields. Friction stir welding (FSW) is an advanced welding process developed by TWI in 1991 [13] and has attracted great attention because of its advantages over conventional fusion welding techniques. These include higher joint efficiency, low distortion, and low residual stress [14,15]. FSP is a solid-state joining process derived from conventional friction welding, in which a non-consumable rotating pin tool is plunged into the filing metal block and the substrate. Frictional heat generated from the tool shoulder and the tool pin completely softens the filling blocks and partially softens the substrates being joined, which are then deformed around the tool [16]. In order to confirm the feasibility of this method, the microstructure, microhardness, and mechanical properties of repaired 34CrNiMo6 steel were analyzed in this study.

## 2. Experimental Procedures

The experiments were performed on an FSP equipment self-modified from a X53K vertical milling machine (Nantong Machine Tool Co., Ltd., Nantong, China), which consisted of a 1600 mm long and 400 mm wide movable worktable (traverse speed from 23.5 mm/min to 1180 mm/min) and a rotating shaft with adjustable rotation speed (from 37.5 rpm to 1500 rpm). Additionally, the tilt angle of the rotating shaft could be adjusted. Argon gas was applied to prevent the weld zone from oxidization and contamination during FSP. When welding high melting point metal, the tool with a screw pin was easy to plug up [17]. Therefore, a tool with a screw-free pin was used in this experiment. The size of the tool is shown in Figure 1, and the friction stir process is shown in Figure 2. As shown in Figure 2, because of argon protection, the metal was silver-white after welding, the tool was at a high temperature, and the toe flash on the advancing side (AS) was less than the on the retreating side (RS). The dimension of the unrepaired sample is shown in Figure 3. The filling block acquired from 34CrNiMo6 forgings corresponding to the trapezoidal groove size was machined to repair the samples. The substrate and the filling block surface were ground with SiC paper and cleaned with acetone prior to FSP repair. The filling block was fixed to the substrate by means of electron beam welding. The chemical composition (wt %) of 34CrNiMo6 steel is reported in Table 1. The welding point position is shown in Figure 3. The processing parameters are listed in Table 2.

An Omniscan MX2 phased array flaw detector manufactured by Olympus was used for flaw detection. Defect-free samples were machined for the standard bar tensile test. Considering the actual use of the repaired components, the tensile specimen contained a part of the substrate. The shape and the size of these samples are shown in Figure 4. The tensile test was carried out by an electronic universal tensile tester with a tensile rate of 0.5 mm/min. The microhardness of the cross section of the repaired specimen was measured by a Qness A10+ microhardness tester (Qness GmbH, Salzburg, Australia). The repaired samples for microstructure observation were machined through a wire electrolytic discharge machine (Jiangnan Wire Cutting Co., Ltd., Nanchang, China) and etched with a 4% nitric acid alcohol solution for 10 s. The microstructure and phase compositions of the repaired samples were examined with Nova Nano SEM450 FE SEM (FEI, Hillsboro, OR, USA) and Zeiss Axis Scope A1 (Zeiss, Jena, Germany). The fracture morphology was also examined with SEM. The XRD analysis was performed using a Bruker D8 advance A25 X-ray diffractometer (Bruker, Berlin, Germany) with monochromatic CuKα radiation operated at 40 KV and 40 mA. X-ray diffraction (XRD) analyses were carried out to confirm the phase structure of the stirred zone (SZ).

## 3. Results and Discussion

### 3.1. Microstructure

Figure 5 shows the results of ultrasonic phased array detection which used the C scan as the scan mode. The signal of a 0.1*ϕ* hole defect was defined as the standard and is shown in Figure 5a. Figure 5b shows the signal detected for the repaired sample, which was recorded at the back of the sample. Figure 6a shows the surface of the FSP-repaired sample. The surface of the repaired area was silver-white. The repaired zone was well protected from oxidation by an argon shield. Figure 2b shows the actual repair process. The color of the repair process was bright white. It can be concluded that the temperature was higher than 1000 °C and less than that corresponding to the melting point in the repair process.

Figure 6b shows the cross-section features of the repaired sample, which contained the repair zone and the substrate. The repair zone included the SZ, the thermo-mechanically affected zone (TMAZ), and the heat-affected zone (HAZ). In order to describe the microstructure more clearly, the sample was divided into nine zones. Zones A1–A3 were at the bottom of the sample: A1 was the HAZ near the substrate, A2 was near the interface between the TMAZ and the HAZ, and A3 was the TMAZ at the bottom of the sample; B1–B3 were the SZ: B1 was near the original interface (about 0.5 mm), and B3 was near the surface (about 0.1 mm); C1–C3 were at the upper left of the original interface: C1 was between the SZ and the TMAZ, C2 was in the TMAZ, and C3 was near the interface between the TMAZ and the HAZ.

As is shown in Figure 7d–f, a high-volume fraction of lath martensite, low bainite, and some retained austenite was found in the SZ. The size of the martensite was different in different positions of the SZ. Although no precise thermometer was used, the temperature of the stirring zone could be roughly determined by the color of the tool. As shown in Figure 2b, it can be speculated that the temperature of the SZ was much higher than the austenite transformation temperature (Ac3). This conclusion was also confirmed by the microstructure found in the zone. The heat input in the SZ came mainly from the friction between the pin tool and the workpiece, which was also the main source of heat for FSP. At the same time, because of the close distance to the shoulder affected zone, the top region of the SZ was subject to more heat generated by the friction between the shoulder and the workpiece surface. Under the combined action of temperature and deformation, austenite transformation occurred in the SZ. Because the SZ was the region with the largest heat accumulation and the cooling rate was relatively stable, the austenite had enough time to transform into martensite. When the austenite was subcooled to a temperature range below the pearlite transformation temperature and above the martensite transformation temperature, bainite transformation occurred from shear transformation to short-range diffusion. Bainite transformation was also affected by the heat input.

As shown in Figure 7a, a large number of retained austenite island (A_R_) were found in the HAZ. As previously analyzed, deformation and heat input caused the phase transition. However, unlike the SZ, the heat dissipation rate at the bottom of the plate was faster, because the bottom of the plate was directly in contact with the clamp, and the austenite produced by deformation-induced transformation did not have enough time to transform. Comparing Figure 7b with Figure 7c, the sample in Figure 7b was found to contain a larger martensitic fraction, and no trace of retained austenite was observed. Figure 7b shows an area closer to the SZ than Figure 7c, where there was enough time for martensite transformation.

As shown in Figure 7g–i, the most complex microstructures were found in this area. In the upper part of the onion rings, there were bands at the junction between the SZ and the HAZ (Figure 6 C2). Table 3 shows the EDS analysis results of the banded structure of the SZ on the repaired sample. It was found that there was almost no difference in composition between the three regions. Furthermore, to better understand the microstructure evolution of the FSW welded regions, the EDS elemental map was examined to identify localized compositional differences in more depth [18]. It could be concluded that the existence of bands, which were predominantly martensitic, was due to the inconsistency of plastic metal flow velocity between the AS (advancing side) and the BS (returning side) during friction stir welding and the difference in velocity between the two grain bands. The same phenomenon was found by Ueji et al. [19]. A high-volume fraction of lath martensite, small amount of pearlite, and A_R_ (Retained austenite) were found, as shown in Figure 7g. Figure 7i shows a band consisting of coarsening lath martensite. The size of martensite in this band was larger than in the other zone. The reason for the formation of coarse martensite is thermomechanical processing. Thermomechanical processing can cause large deformation which can promote the transformation of austenite into ferrite [20].

In summary, the evolution of the microstructures was caused mainly by two factors. One of the factors was the heat input. During the processing, the microstructure of the SZ and TMAZ transformed into austenite due to the influence of the heat input [21]. As the tool pin moved, the repaired zone began to cool. Different cooling rates led to different microstructure transformations. The second factor was the deformation. The process caused a large deformation which promoted austenite transformation into ferrite. At the same time, the size of the martensite was also affected. In order to confirm the analysis of the microstructure, crystal phases of the SZ and the substrates were identified by XRD. The XRD patterns are presented in Figure 8, which shows α_M_ (α-Fe in martensite) arising from different hkl planes, with varying intensities. The bainite peaks are attributable to α (110) and α (200) in the SZ and the substrates [7]. The austenite reflections in the substrate sample derived from (111), (200), and (220). No austenite peaks were found in the SZ samples. Combining the results of XRD, SEM, and OM, it could be confirmed that the SZ was composed of martensite and bainite, without austenite.

### 3.2. Mechanical Properties

Figure 9 shows the microhardness distribution for the repaired 34CrNiMo6 steel. The microhardness in the repaired zone was much higher than in the substrates. At the same time, the microhardness values showed differences in the retreating side and advancing side. The mean microhardness value in the SZ was about 650 HV. As analyzed in Section 3.1, a high-volume fraction of bainite and lath martensite was found in the SZ. At the same time, the stir zone had a refined grain size due to the effect of the tool pin. The microhardness in the TMAZ was 550–620 HV. It was found that the hardness value fluctuated greatly in the TMAZ. Big fluctuations of microhardness were found in the AS. The microstructure in the TMAZ was the most complex. Both quenched martensite and tempered martensite were found in the zone. The formation of pearlite could affect the microhardness of the zone. The tool pin showed a lower effect on the TMAZ than on the SZ, so grain refinement was better than in the SZ. At the same time, the grains were elongated. The microhardness of the HAZ was much lower than those of the SZ and TMAZ, whose value was about 360–420 HV. The microhardness showed a downward trend in the HAZ of the AS, while it fluctuated smoothly in the HAZ of the RS. The microhardness was only 7% lower than in the substrate, which was much better than the result obtained with the fusion repair method. The different heat input in the AS and RS could explain this phenomenon. The AS often generated more heat than the RS. Compared with the SZ and TMAZ, the HAZ was only affected by the heat input. The heat input of FSP is much lower than that of conventional fusion welding. Therefore, grain growth is not severe, and the HAZ is small.

The room temperature tensile properties of the repaired specimens are listed in Table 4. It was found that the maximum ultimate strength (UTS) was 900 MPa and could reach 91.8% of that of the substrate (the maximum ultimate strength of the substrate was 980 MPa [22]). The mean ultimate strength was 883 MPa, which was about 90% of that of the substrate. Three tensile specimens were fractured in HAZ. The results of the microstructural analysis indicated that a high-volume fraction of lath martensite formed in the SZ. The results of the microhardness and the XRD results could also confirm the microstructure of the SZ. The high strength can be attributed to the microstructure of the high-volume fraction of lath martensite. The microhardness results of the retreating side indicated that the values of this side were the lowest, which could cause fracture in the specimen. To confirm the fracture position of the tensile specimen, a cross-sectional photo of the fractured tensile specimens was analyzed (Figure 10). It was found that the fracture mainly crossed through the HAZ. The optical image and SEM image of the fractured tensile specimen were also used to analyze the microstructure. The microstructure of the HAZ was composed of ferrite and pearlite. In the former microstructure analysis, the SZ consisted of martensite and bainite, while the fracture position consisted of ferrite and pearlite. It is well known that martensite and bainite have high strength, while ferrite and pearlite have low strength. Therefore, the fracture occurred in the HAZ, which is in accordance with the results of microhardness. In order to better understand the fracture mechanism, the fracture morphology was analyzed. Figure 11 shows the SEM image of fracture morphology in specimen No. 2. It should be noted here that, as Figure 10 shows, a heavy necking took place during tensile loading. The most complex fracture morphology found is shown in Figure 11. The typical ductile ‘cup-and-cone’ fracture surface included three parts: the fiber region in the center, the outside smaller shear lip, and the larger radial region between the two [23]. Although the fracture morphology in Figure 11 does not conform to the typical ductile ‘cup-and-cone’ fracture surface, some ductile characteristics were found in the fracture morphology. Figure 11b shows some elongated shapes of voids, which may indicate that localized shear stresses also occurred, in addition to tensile stresses, during deformation. The results of the tensile tests were in accordance with the microstructure analysis. Both dimples and small steps were found in the fracture surface. The fracture mode of the tensile samples was ductile/brittle mixed fracture.

## 4. Conclusions

We demonstrated that: 

(1) A 34CrNiMo6 filling block was successfully used to repair the 34CrNiMo6 substrate with no defects. The 34CrNiMo6 filling block was metallurgically combined with the substrate well. 

(2) The microstructure in the SZ was lath martensite in high volume and small amounts of austenite and bainite. At the bottom of the samples, a large amount of retained austenite was found. The austenite transformed into martensite at the TMAZ of the bottom. In the upper left of the samples, many banded structures consisting of martensite were found.

(3) The mean microhardness value was about 650 HV and was due to the high-volume fraction of lath martensite. The microhardness values (550–620 HV) lightly decreased in the TMAZ, due to the formation of A_R_. The microhardness of the HAZ was much lower than that of the SZ and TMAZ, whose value was about 360–420 HV.

(4) The maximum ultimate strength was 900 MPa and could reach 91.8% of the maximum ultimate strength of the substrate. The fracture mode of the tensile samples was ductile/brittle mixed fracture.

## Figures and Tables

**Figure 1 materials-12-00279-f001:**
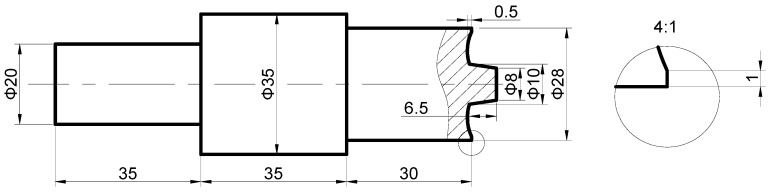
Photograph showing the dimensions of the tool.

**Figure 2 materials-12-00279-f002:**
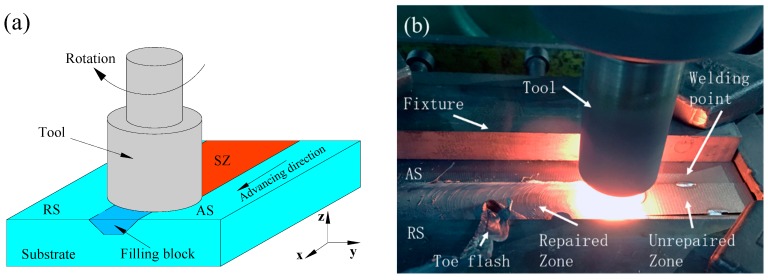
Schematic image of friction stir processing (FSP) repair: (**a**) schematic image, (**b**) actual process. RS, retreating side, AS, advancing side, SZ, stirred zone.

**Figure 3 materials-12-00279-f003:**
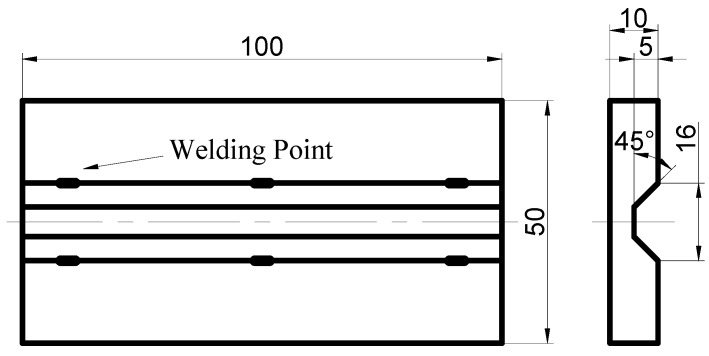
Geometric shape and size of the unrepaired plate.

**Figure 4 materials-12-00279-f004:**
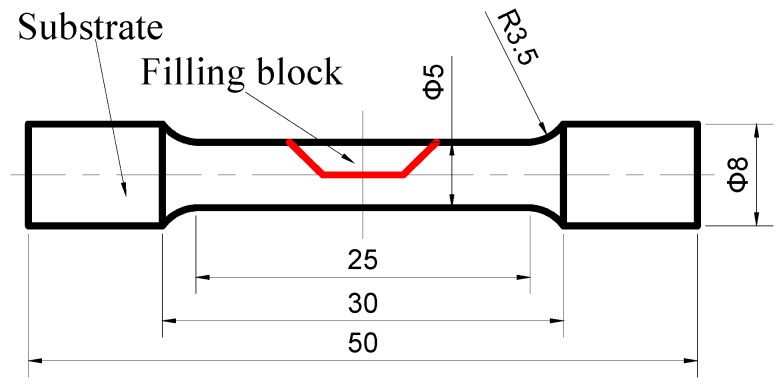
Geometric shape and size of the room-temperature tensile specimen.

**Figure 5 materials-12-00279-f005:**
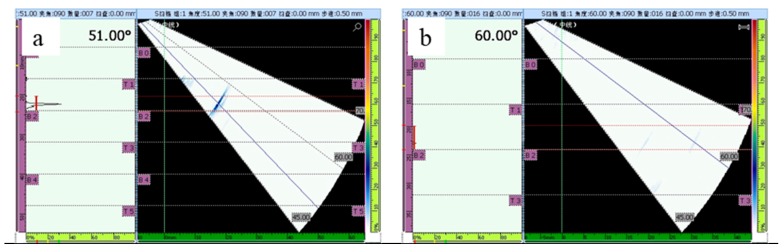
Results of ultrasonic phased array detection: (**a**) photo of the signal detected for a defect consisting of a 0.1*ϕ* hole, (**b**) photo of the signal detected for the repaired sample.

**Figure 6 materials-12-00279-f006:**
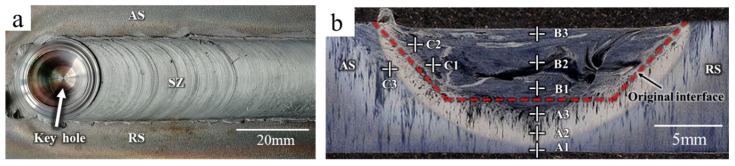
Surface and cross section features of the FSP-repaired samples: (**a**) photo of the repaired sample surface, (**b**) cross-sectional morphology of the repaired sample.

**Figure 7 materials-12-00279-f007:**
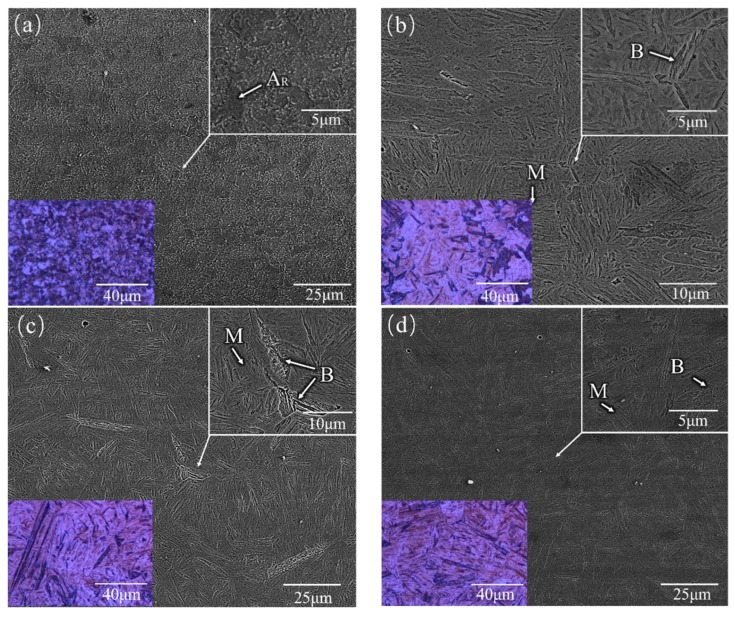
Typical SEM photos of FSP-repaired 34CrNiMo6 shown in Figure 5 (small box for partial enlargement): (**a**–**c**) zones A1–A3; (**d**–**f**) zones B1–B3; (**g**–**i**) zones C1–C3; (**j**) substrate. The color photos are optical microscope (OM) photos (A_R_: Retained austenite; M: Martensite; B: Bainite; P: Pearlite).

**Figure 8 materials-12-00279-f008:**
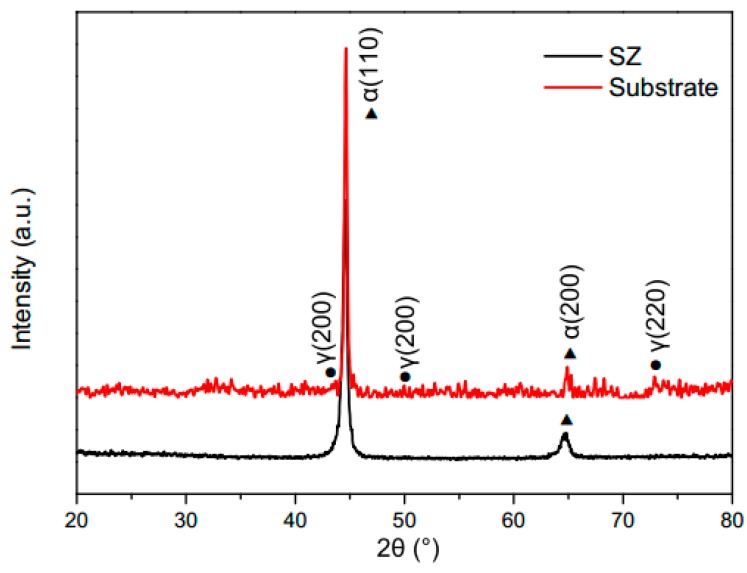
XRD patterns of the SZ of the FSP-repaired specimen.

**Figure 9 materials-12-00279-f009:**
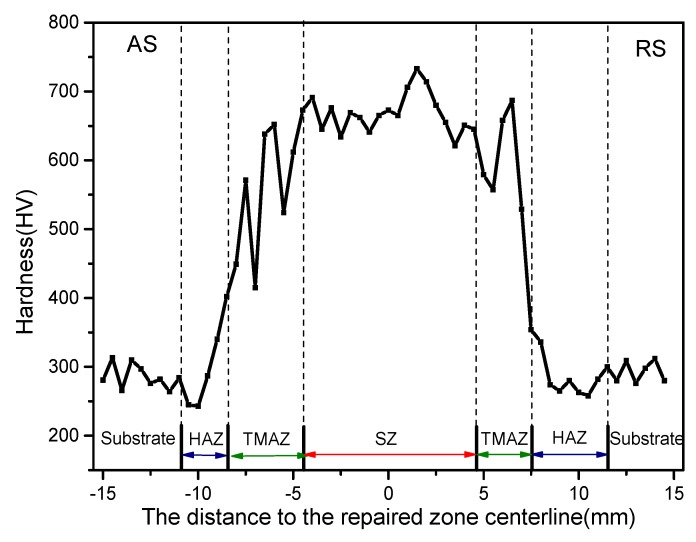
Microhardness distribution for FSP-repaired 34CrNiMo6. HAZ, heat-affected zone, TMAZ, thermo-mechanically affected zone.

**Figure 10 materials-12-00279-f010:**
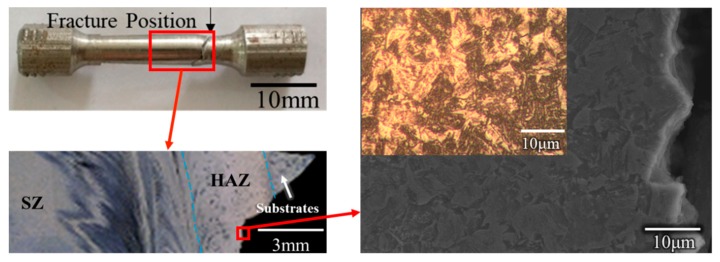
Image of the fracture position in the tensile specimens.

**Figure 11 materials-12-00279-f011:**
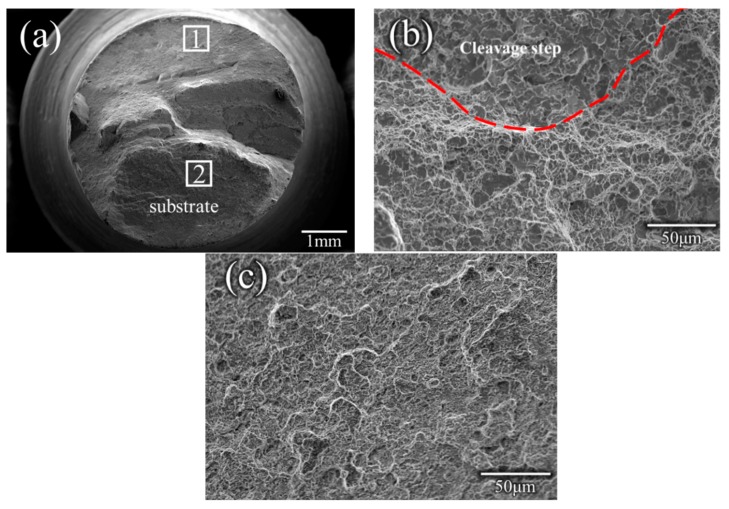
SEM image of the fracture morphology in a specimen, (**a**) Macroscopic appearance of the fracture (**b**,**c**) higher magnification SEM image of region 1 and region 2.

**Table 1 materials-12-00279-t001:** Alloying composition of 34CrNiMo6 alloy (wt %).

Element	C	Cr	Ni	Mo	Mn	Si	Fe
Content	0.34	1.5	1.5	0.25	0.50	0.40	Balance

**Table 2 materials-12-00279-t002:** Processing parameters of FSP-repaired 34CrNiMo6 alloy.

Rotation Speed (rpm)	Traverse Speed (mm/min)	Tilt Angle (°)	Argon Gas Rates (L/min)
600	47.5	3	20

**Table 3 materials-12-00279-t003:** EDS analysis results of the banded structure of the SZ in the repaired sample (regions 1–3 are the region marked in Figure 7h) (wt %).

Element	Fe	Cr
Region 1	98.64	1.36
Region 2	98.84	1.16
Region 3	98.74	1.26

**Table 4 materials-12-00279-t004:** Tensile testing results of the repaired sample.

Sample	UTS (MPa)	Fracture Position
1	860	HAZ
2	900	HAZ
3	890	HAZ

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
