# Peer review of "Microstructure and Mechanical Properties of 34CrNiMo6 Steel Repaired by Friction Stir Processing"

_materials, 2019, doi:10.3390/ma12020279_

Round 1
Reviewer 1 Report
This paper is a report of the microstructural study and mechanical properties during a repairing FSP method for the 34CrNiMo6 steel structure. The work aims to explain the correlation between the microstructure and mechanical properties –specifically the tensile strength (pages 8 and 9). It seems basically worth to be accepted in its present form. However, regarding the principles of physical metallurgy, authors should consider the following comments.
COMMENT 1. [Page 7, Line 151] the EDS results for the banded structure has been reported. To present a firm interpretation for the composition of the bonded layers, an EDS elemental map is also required. If authors are reporting the point EDS analysis, it should be mentioned that the elemental map also can elucidate the composition of the banded layers or other regions of the FSW weld (HAZ, TMAZ, SZ).
This reviewer suggests it be added to the manuscript to complete the EDS interpretation –Page 7, Line 152-:
“Furthermore, for some more development in the microstructure evolution of the FSW weld regions, the EDS elemental map can also reveal the localized compositional differences in more depth [Ref.*].
[Ref.*]: Tamadon, A.; Pons, D.J.; Sued, K.; Clucas, D. Thermomechanical Grain Refinement in AA6082-T6 Thin Plates under Bobbin Friction Stir Welding. Metals 2018, 8, 375. https://doi.org/10.3390/met8060375
COMMENT 2. [Paragraph 2, Page 7] Authors mentioned that the crystal phase of the substrate and SZ were identified by XRD (figure 8), but figure 8 only illustrates the XRD pattern of the SZ. For a comparison with the substrate, authors should analyze the substrate and report the XRD pattern to show the differences between the substrate and the SZ. Or, authors can report the XRD characterization results of the substrate (peaks) from a reference and cite it in their work (as a comparison with the analyzed SZ results).
Author Response
Point 1. [Page 7, Line 151] the EDS results for the banded structure has been reported. To present a firm interpretation for the composition of the bonded layers, an EDS elemental map is also required. If authors are reporting the point EDS analysis, it should be mentioned that the elemental map also can elucidate the composition of the banded layers or other regions of the FSW weld (HAZ, TMAZ, SZ). This reviewer suggests it be added to the manuscript to complete the EDS interpretation –Page 7, Line 152-: “Furthermore, for some more development in the microstructure evolution of the FSW weld regions, the EDS elemental map can also reveal the localized compositional differences in more depth [Ref.*]. [Ref.*]: Tamadon, A.; Pons, D.J.; Sued, K.; Clucas, D. Thermomechanical Grain Refinement in AA6082-T6 Thin Plates under Bobbin Friction Stir Welding. Metals 2018, 8, 375. https://doi.org/10.3390/met8060375
Response1: The point EDS analysis is reported in this paper. This point EDS analysis report is to prove that the composition of banded structure is consistent with the other region. This report is also to prove that the formation of banded structure is a trace of plastic flow rather than composition differences. This phenomenon is also founded in REF [19]. The [Ref.*] mentioned by the reviewer is also added in this paper.
Point 2. [Paragraph 2, Page 7] Authors mentioned that the crystal phase of the substrate and SZ were identified by XRD (figure 8), but figure 8 only illustrates the XRD pattern of the SZ. For a comparison with the substrate, authors should analyze the substrate and report the XRD pattern to show the differences between the substrate and the SZ. Or, authors can report the XRD characterization results of the substrate (peaks) from a reference and cite it in their work (as a comparison with the analyzed SZ results).
Response2: The XRD pattern of the substrate is added in this paper.
Reviewer 2 Report
This work reports on a using friction stir processing as a means to repair 34CrNiMo6 alloys, using a block of the same material cut to the shape of the groove as the filler material. It is shown that good chemical bonding has been achieved. In tensile tests, samples failed at HAZ; while reaching more than 90% of the base material's strength.
The report is interesting and the results are promising; however, there are issues both with the text and experimental that need attention and revision before the paper is accepted for publication.
The abstract is not clearly written and needs to be revised. Those parts that require revision are marked in the attached document. In general the manuscript suffers from poor English in some parts. Please refer to the attached documents for more information.
The characterization results also need improvement. For example in Fig 7, the SEM images are not clear. They need to be enhanced for brightness and contrast. In general, in analyzing the weld microstructures, the authors have mostly relied on SEM images. There is level of uncertainty for the microstructure and phase identification. SEM images cannot be interpreted freely. They have to be combined with optical metallography images, EBSD and even TEM to be able to identify different phases and microstructures with more certainty. Moreover, in making conclusions regarding the microstructural evolution during joining, assumptions have made regarding the temperature regime. It seems necessary for this study to use thermocouples to measure temperature in different zones to make sure what areas reach Ac3 temperature.
Also, in analyzing the banded structure in the stir zone, the authors have ruled out the variation of composition within the bands, relying on EDS results. The small variations of composition (beyond the detection limit and spatial resolution of the EDS in SEM) could contribute to variation in microstructure between different bands. The last paragraph in page 6 and the first paragraph of page 7 need to be revised.
Again, the authors are encouraged to go through all the comments and mark-ups in the attached documents carefully and apply changes accordingly.

Author Response
Point 1: The abstract is not clearly written and needs to be revised. Those parts that require revision are marked in the attached document. In general the manuscript suffers from poor English in some parts. Please refer to the attached documents for more information.
Response1: The abstract has been revised according to the expert opinions. And the language has been revised according to the attached documents.
Point 2: The characterization results also need improvement. For example in Fig 7, the SEM images are not clear. They need to be enhanced for brightness and contrast.
Response2: The Fig 7 has been enhanced for brightness and contrast.
Point 3: In general, in analyzing the weld microstructures, the authors have mostly relied on SEM images. There is level of uncertainty for the microstructure and phase identification. SEM images cannot be interpreted freely. They have to be combined with optical metallography images, EBSD and even TEM to be able to identify different phases and microstructures with more certainty.
Response3: The optical metallography images are provided to confirm the SEM results.
Point 4: Moreover, in making conclusions regarding the microstructural evolution during joining, assumptions have made regarding the temperature regime. It seems necessary for this study to use thermocouples to measure temperature in different zones to make sure what areas reach Ac3 temperature.
Response4: Thanks for the advice. The thermocouples were not used in this experiment. The temperature assumptions are based on the REF [17] and the color of the tool showed in Fig.2b. At the same time, the temperature can be also inferred according to the microstructure evaluation. In the discussion of REF [17], the temperature in the weld was analyzed. While no melting occurs during FSW, very high temperature was measured, certainly in excess of the transformation temperature.
Point 5: Also, in analyzing the banded structure in the stir zone, the authors have ruled out the variation of composition within the bands, relying on EDS results. The small variations of composition (beyond the detection limit and spatial resolution of the EDS in SEM) could contribute to variation in microstructure between different bands.
Response5: The EDS is only to confirm that the composition of the bands is in consistence with the other region. About the bands, the REF [19] has explained the phenomenon. This bands are traces of plastic flow.
Point 6: The last paragraph in page 6 and the first paragraph of page 7 need to be revised.
Response6: The last paragraph in page 6 and the first paragraph of page 7 has revised according to the expert opinions. And I would like to explain these sentences in this paragraph (It can be concluded that the existence of bands which are predominantly martensitic are due to the inconsistency of plastic metal flow velocity between the AS and the BS during friction stir welding, and the difference in velocity between the two grain bands.) Plastic flow will occur in this process. And the grain bands are formed in the process. The banded structure is the trace of grain bands flow.
Reviewer 3 Report
This study contains interested results; this study could be further improved if residual stresses measurement is included or also post-welding heat treatment.
Author Response
Point1: This study contains interested results; this study could be further improved if residual stresses measurement is included or also post-welding heat treatment.
Response: Thanks for the advice provided by the experts. The residual measurement and post-welding heat treatment will carry out in the follow-up study.